# Synthetic Routes to 2-aryl-1*H*-pyrrolo[2,3-*b*]pyridin-4-amines: Cross-Coupling and Challenges in SEM-Deprotection

**DOI:** 10.3390/molecules29194743

**Published:** 2024-10-07

**Authors:** Srinivas Reddy Merugu, Sigrid Selmer-Olsen, Camilla Johansen Kaada, Eirik Sundby, Bård Helge Hoff

**Affiliations:** 1Department of Chemistry, Norwegian University of Science and Technology (NTNU), N-7491 Trondheim, Norway; srinivas.r.merugu@ntnu.no (S.R.M.); sigrid.selmer-olsen@osloskolen.no (S.S.-O.); johansencam@gmail.com (C.J.K.); 2Department of Materials Science and Engineering, Norwegian University of Science and Technology (NTNU), N-7491 Trondheim, Norway; eirik.sundby@ntnu.no

**Keywords:** azaindole, chemoselective Suzuki–Miyaura, Buchwald–Hartwig amination, SEM-deprotection, 8-membered 7-azaindole, CSF1R

## Abstract

7-Azaindoles are compounds of considerable medicinal interest. During development of the structure–activity relationship for inhibitors of the colony stimulated factor 1 receptor tyrosine kinase (CSF1R), a specific 2-aryl-1*H*-pyrrolo[2,3-*b*]pyridin-4-amine was needed. Two different synthetic strategies were evaluated, in which the order of the key C-C and C-N cross-coupling steps differed. The best route relied on a chemoselective Suzuki–Miyaura cross-coupling at C-2 on a 2-iodo-4-chloropyrrolopyridine intermediate, and subsequently a Buchwald–Hartwig amination with a secondary amine at C-4. Masking of hydroxyl and pyrroles proved essential to succeed with the latter transformation. The final trimethylsilylethoxymethyl (SEM) deprotection step was challenging, as release of formaldehyde gave rise to different side products, most interestingly a tricyclic eight-membered 7-azaindole. The target 2-aryl-1*H*-pyrrolo[2,3-*b*]pyridin-4-amine (compound **3c**) proved to be 20-fold less potent than the reference inhibitor, confirming the importance of the N-3 in the pyrrolopyrimidine parent compound for efficient CSF1R inhibition.

## 1. Introduction

Azaindoles (pyrrolopyridines) are an important class of heterocycles. Derivatives have been used as probes in biological imaging [1,2], and have a rich coordination chemistry [3]. Most importantly, azaindoles are used as core scaffolds in medicinal chemistry to discovery new bioactivity, and since they are bioisosteres of indoles and pyrrolopyrimidines they are useful probe compounds in structure–activity relationship studies (SAR). The most prominent use of 7-azaindoles is as kinase inhibitors. Depending on their detailed structure, inhibitory activity towards different kinases has been discovered. FDA approved drugs include pexidartinib (PLX3397) [4], a colony stimulating factor 1 receptor kinase (CSF1R) inhibitor for treatment of tenosynovial giant cell tumours, and vemurafenib (PLX4032) [5], a serine/threonine-protein kinase B-Raf (B-Raf) inhibitor for late-stage melanoma (Figure 1). Other interesting kinase targets for 7-azaindoles include Janus kinase 3 (JAK3) [6], ataxia telangiectasia and Rad3-related protein kinase (ATR) [7]; epidermal growth factor receptor kinase (EGFR) [8]; proto-oncogene serine/threonine-protein kinase (PIM) kinase [9], activin-like kinase (Alk5) [10] and calcium/calmodulin dependent protein kinase 2 (CAMKK2) [11]. Methodology for functionalisation of the 7-azaindole scaffold has been provided by Barl et al. [12] and Scheider et al. [13].

By proper planning, intramolecular cyclization can be performed, giving rise to different polycyclic 7-azaindoles. Compound **I** is obtained by a Buchwald–Hartwig amination and a C-H arylation in sequence or tandem [14], while **II**, a fused 7-membered azaindole, is made starting with a Sonogashira coupling and finalising with a palladium catalysed Heck [15]. In contrast, compound **III** and the 8-membered **IV** are obtained after Suzuki–Miyaura cross-coupling followed by Pictet–Spengler condensation with aldehydes [16]. Of relevance to our work is also the tendency of 4-azaindoles to react with formaldehyde giving dimers like **V** [17]. During SAR studies for pyrrolopyrimidine-based CSF1R inhibitors [18,19,20], we needed the corresponding 7-azaindole. Despite a high synthetic activity in this field, routes to similar 2-aryl-1*H*-pyrrolo[2,3-*b*]pyridin-4-amines have not been published. Thus, here we report our investigation into synthetic routes involving protection group selection and implementation of cross-coupling methodology. Moreover, we also identified a new class of tricyclic azaindoles incorporating an 8-membered ring.

## 2. Results

The pyrrolopyrimidine **1** (Figure 1) is a potent CSF1R inhibitor in enzymatic studies [18], while the corresponding thienopyrimidine **2** was far less active, indicating the NH to be important in inhibitor binding to the kinase. To investigate SAR further, the corresponding 7-azaindole analogue **3c** was targeted. We assumed that a similar synthetic plan as that used for the corresponding pyrrolopyrimidines could be viable, involving a thermal S_N_Ar nucleophilic aromatic substitution and a Suzuki–Miyaura cross-coupling; see Figure 1.

### 2.1. Route A: Amination Followed by Suzuki–Miyaura Cross-Coupling

In the first synthetic approach, we evaluated the possibility of starting with an amination reaction at C-4, to give a 2-iodo derivative suitable for Suzuki–Miyaura cross-coupling; see Figure 2. First, the SEM-protected building block **7** was attempted aminated in refluxing *n*-butanol under thermal S_N_Ar conditions. However, after 48 h, no conversion was observed by ^1^H NMR spectroscopy. Noting that previous studies reported rather forcing conditions [21], we instead decided for palladium catalysed amination.

The RuPhos ligand has been found to be useful in similar aminations [22,23]. However, treatment of the 2-iodo derivative **7** under these conditions was unsuccessful and instead led to reduction at C-2 giving **6** and an unstable molecule assumed by ^1^H NMR to be structure **8** (Figure 2). The above results strongly indicated that the oxidative addition of palladium preferably occurred at C-2 instead of at C-4. Therefore, we decided to introduce the 4-amino group prior to iodination. Palladium catalysed amination of 4-chloro-7-azaindole (**4**) with *N*-benzylmethylamine has previously been reported by Henderson et al. with a 96% yield [22], but their ^1^H NMR spectra were not in accordance with the proposed structure. We performed this reaction using the RuPhos ligand and the RuPhos Pd G2 pre-catalyst and achieved a 33% yield of the aminated structure **9**. Loss in mass was seen in the extraction, indicating that water soluble side products were formed. Inferior result and incomplete conversion were noted when performing the same transformation with sodium *tert*-butoxide as base. (Under similar conditions the benzenesulfonyl derivative **5** gave after purification, 20% of the aminated structure **9** and 48% of 4-chloro-7-azaindole (**4**). Due to the low yield and the inconvenient deprotection occurring, we instead evaluated the use of the SEM-protected analogue **6** in conversion to the aminated derivative **10.** The reaction was tested with XPhos Pd G2/XPhos, RuPhos Pd G2/RuPhos and Pd(OAc)_2_/RuPhos in *n*-butanol and *tert*-butanol. Dried solvents are needed for the reaction to proceed with ease. The highest conversion and yield were seen for the use of Pd(OAc)_2_/RuPhos in *tert*-butanol, giving 94% conversion in 5 min and 68% isolated yield. Reduction at C-4 leading to **11** (6%) was observed as a side reaction. This transformation can very likely be improved by further tuning of conditions. Unfortunately, the iodination of **10** attempted with 1.5 equiv. of lithium diisopropylamide (LDA) only resulted in 8% conversion to product and 6% isolated yield. The same reaction ran with 3 equiv. of LDA, terminated at 12% conversion. The use of *n*-butyl lithium could have led to better results. This was however not tested. Suzuki–Miyaura cross-coupling on **12** with XPhos Pd G2/XPhos provided the 2-phenyl derivative **13a** with 77%. While **13a** could be deprotected (see Section 2.3), the route proved inefficient due to a difficult iodination.

### 2.2. Route B: Suzuki–Miyaura Cross-Coupling Followed by Amination

In the alternative approach, we aimed to establish a chemoselective Suzuki–Miyaura cross-coupling at C-2 on the 4-chloro-2-iodo intermediate **7**. A similar reaction has been performed on the corresponding methyl [9] and benzenesulfonyl *N*-1 protected analogues [8]. The initial experiments were based on our previous selectivity study on thienopyrimidines [24], that showed preference for mono arylation at C-6 (analogous to C-2 in pyrrolopyridines) when using Pd(OAc)_2_ or tris(dibenzylideneacetone)dipalladium (Pd_2_(dba)_3_). Both these systems were investigated in a model reaction with 4-methoxyphenyl boronic acid, of which Pd_2_(dba)_3_ showed excellent selectivity for C-2 arylation, giving the *para*-methoxy compound **14b** 68–71% isolated yield at 0.5–1 g scale; see Figure 3. A trace amount of the 2,4-diarylated product **15b** was also observed. The corresponding phenyl derivative **14a** was formed in similar yield (75%).

This catalyst system was then tested for preparation of the *para*-hydroxymethyl derivative **14c**, see Table 1. The cross-coupling provided a 92/4 ratio of mono- to di-arylated product, no reduction product (**6**), but also gave 4% of an unknown side product difficult to remove by silica-gel column chromatography. Lowering the temperature only increased the amount of the problematic impurity (entries 1–3). The effect of varying the catalyst in this transformation was therefore investigated through screening reactions. Aminations catalysed by XPhos Pd G2, XPhos, Pd(OAc)_2_, PEPPSI^TM^-SIPr and (Dppf)PdCl_2_ (entries 4–7) suffered from low chemoselectivity, or the formation of other side product. On the other hand, the use of Pd(PPh_3_)_4_ only led to 5–8% of the C-2 reduced derivative **6** (entry 8). A preparative 1 g reaction (entry 9) resulted in 83% isolated yield of the product **14c**. As the cross-coupling appeared to slow down on scaling, the 2 g reaction (entry 10) was performed at 90 °C for 22 h. Under these conditions, 6% of the diarylated product **15c** was seen, but the product **14c** could anyhow be isolated with 83% yield. Similarly, we prepared the *tert*-butyldimethylsilane (TBMDS) analogue **14d** by Suzuki–Miyaura cross-coupling using Pd(PPh_3_)_4_ with 90% isolated yield (Figure 4). The corresponding SEM analogue **14e** was made from the alcohol **14c** by a substitution reaction in DMF in mediocre 51% isolated yield. The formyl ester **14f** was also isolated as a side product, originating from a reaction between the formed alkoxide and DMF (Figure 4).

The next step was a palladium mediated amination of the advanced intermediates **14**. We first aminated the phenyl and methoxy derivatives **14a–b** using Pd(OAc)_2_ and the RuPhos ligand. Encouragingly, both substrates were fully converted in 1 h, giving the aminated products **13a–b** with 74% and 76% isolated yield, respectively (Table 2, entries 1–2). However, when attempting the same reaction with the unprotected **14c**, less than 40% conversion was seen in 24 h (Table 2, entry 3) and the product **13c** could not be isolated in sufficiently pure form. Four ligands/catalyst systems including Pd(OAc)_2_/XantPhos PEPPSI^TM^-SIPr pre-catalyst, tri(*o*-tolyl)-phosphine and (Dppf)PdCl_2_ were tested without any improvement in reaction outcome. Clearly, the acidic hydroxyl proton was again causing side reactions to dominate. The palladium catalysed amination of the TBDMS derivative **14d** proceeded with good conversion. Some deprotection of the hydroxymethyl group also giving **13c** was observed if the reaction was left stirring for several h. However, quenching the reaction after only 0.5 h allowed for isolation of the TBDMS protected analogue **13d** with an 89% yield. The double SEM-protected analogue **14e** was smoothly aminated to **13e** (entry 5).

### 2.3. SEM-Deprotection

Regarding synthesis of the corresponding 6-arylated pyrrolopyrimidines, most SEM-deprotections proceeded with ease [19,25] by a two-step procedure, involving first the trifluoroacetic acid (TFA) treatment followed by a basic step. It has been assumed that the acidic step leads to a mixture of the product **3** and the intermediate **VI**. Then, basic conditions release formaldehyde and give full conversion to **3** (Figure 5).

As the precursor **13c** was difficult to obtain using Buchwald–Hartwig amination, our main target, compound **3c,** could hopefully be formed by deprotection of **13d** or **13e**. The double deprotection of the *tert*-butyldimethylsilyl protected **13d** proceeded decently and the product **3c** was isolated with 51% yield after one round of silica-gel column chromatography and a crystallisation. The crystallisation ensured high quality material for the biotesting. In contrast, for the reaction of the double SEM protected analogue **13e**, the O-SEM group at the benzylic position was less reactive, and a high number of compounds were observed. Thus, a low 13% yield was obtained after column chromatography. Interestingly, the 8-membered azaindole **16c** was also isolated. The compound is evidenced by its mass, and a characteristic methylene unit residing at ca 31 ppm, with HMBC NMR coupling both to the phenyl unit of the 4-amino group and the pyrrole ring. Obviously, during SEM-deprotection formaldehyde is released which under the given conditions undergoes a two-step electrophilic aromatic substitution. The use of BF_3_-OEt_2_ in the deprotection of **13d–e** was also attempted, but this resulted in even more complex reaction mixtures. Tetrabutylammonium fluoride was not evaluated.

Triggered by the interest for this new tricyclic eight-membered azaindole, we also performed deprotection of the analogue **13a**. Performing the acidic step at 22 °C (rt) gave the product **3a** with a 45% yield, alongside multiple products which we did not manage to isolate. When the same reaction was repeated at 50 °C, **3a** was isolated with a 31% yield alongside 12% of the tricyclic 8-membered azaindole **16a**. To increase the amount of **16a**, we stirred **3a** with formaldehyde (1.2 equiv.) and TFA. The main product under these conditions, however, was the dimer **17a** (32%). The most characteristic ^13^C NMR signal is from the central methylene unit having a carbon shift of 23.8 ppm, distinguishing this structure from **16a**. Although similar azaindole dimers have been made [17,26], this dimer is much more sterically crowded than those reported. Then, the methoxy analogue **13b** was deprotected. After 6 h at rt with TFA and an extended stirring time with aq. NaHCO_3_, the product **3b** was isolated after two rounds of column chromatography with a low 14% yield (semi-pure). We assumed that one of the side products was the 8-membered azaindole **16b**. To promote formation of this compound, we reasoned that longer reaction time in TFA could be beneficial. Thus, the TFA treatment step was performed for 18 h instead of 6 h at 50 °C. This gave **16b** as the main product with 69% isolated yield. Similar yield was obtained when reacting at 22 °C, while the addition of more formaldehyde had a negative effect (more complex reaction mixture).

### 2.4. Enzymatic CSF1R Kinase Activity

The initial aim of these synthetic endeavours was to evaluate the azaindole **3c** as a CSF1R inhibitor. The CSF1R kinase inhibitory activity of **3c** was compared with that of the corresponding pyrrolopyrimidine **1** and the thienopyrimidine **2**. IC_50_ titration at low ATP concentration indicated that the azaindole **3c** was decently active (IC_50_ = 3.0 nM), see Figure 2.

However, when profiled in an alternative CSF1R assay with higher ATP concentration, the IC_50_ value was much higher (IC_50_ = 105 nM), and compared to the corresponding pyrrolopyrimidine 20-fold lower activity was seen (Table 3). Apparently, the inhibitory profile of **3c** is very sensitive to the ATP concentration. Obviously, the N-3 nitrogen in the pyrrolopyrimidines is of importance. The loss in activity is also in line with the X-ray co-crystal of the pyrrolopyrimidine **1** with the kinase domain of CSF1R, where N-3 is involved in hydrogen binding via a deeply situated water molecule [19].

## 3. Materials and Methods

### 3.1. Chemicals and Analysis

4-Chloro-7-azaindole was purchased from 1Click Chemistry. All other reagents, starting materials, palladium catalysts and solvents were purchased from Sigma-Aldrich and used as is. Dry solvents were collected from a Braun MB SPS-800 solvent purification system. Reactions were monitored by thin-layer chromatography (TLC) using silica-gel on aluminium plates, F254, Merck. Purification of compounds by flash column chromatography was performed with silica-gel (40–63 mesh, 60 Å) using standard glassware. NMR spectra were recorded on a Bruker Avance III HD 400 or 600 MHz instrument in either CDCl_3_ containing tetramethylsilane or DMSO-*d*_6_ as solvents. ^1^H and ^13^C chemical shifts are reported in part per million (ppm) using tetramethylsilane (0.00 ppm) or residual solvent (DMSO-*d*_5_, 2.50/39.52 ppm) as an internal reference standard. Infrared absorption spectra were recorded on a Thermo Nicolet Nexus FT-IR spectrometer using a Smart Endurance reflection cell. Absorption bands are reported as strong (s), medium (m) or weak (w). Accurate mass determination was performed on a Synapt G2-S Q-TOF instrument from Waters TM in either positive or negative mode. The samples were ionized with an ASAP (APCI) or ESI probe. Exact mass calculations and spectra processing were done using Waters TM Software Masslynx v4.1 SCN871.

### 3.2. Synthesis

#### 3.2.1. 4-Chloro-2-phenyl-1-((2-(trimethylsilyl)ethoxy)methyl)-1*H*-pyrrolo[2,3-*b*]pyridine (**14a**)

To a mixture of 4-chloro-2-iodo-1-((2-(trimethylsilyl)ethoxy)methyl)-1*H*-pyrrolo[2,3-*b*]pyridine (**7**) (350 mg, 0.856 mmol), phenylboronic acid (125 mg, 1.02 mmol), Pd_2_(dba)_3_ (24 mg, 0.026 mmol) and K_2_CO_3_ (350 mg, 2.56 mmol) was added de-gassed 1,4-dioxane:water (1:1, 5 mL) under an N_2_ atmosphere. The reaction mixture was stirred at 100 °C for 30 min before being allowed to cool to rt. The solvent was removed in vacuo, and the residue was partitioned between EtOAc (20 mL) and water (20 mL). The layers were separated, and the aqueous layer was extracted with EtOAc (3 × 20 mL). The combined organic phases were washed with brine (20 mL), dried over anhydrous Na_2_SO_4_, filtered and concentrated in vacuo. The product was purified by silica-gel column chromatography (*n*-hexane/EtOAc, 9:1, R_f_ = 0.30) to give 230 mg (0.641 mmol, 75%) of a yellow oil. ^1^H NMR (600 MHz, CDCl_3_) *δ*: 8.22 (d, *J* = 5.1 Hz, 1H), 7.80–7.78 (m, 2H), 7.50–7.41 (m, 3H), 7.15 (d, *J* = 8.5 Hz, 1H), 6.69 (s, 1H), 5.66 (s, 2H), 3.72 (t, *J* = 8.5 Hz, 2H), 0.95 (t, *J* = 8.5 Hz, 2H), 0.04 (s, 9H); ^13^CNMR (151 MHz, CDCl_3_): 150.4, 143.3, 142.9, 135.4, 131.7, 129.5, 128.9 (2C), 128.8 (2C), 120.0, 117.1, 99.2, 71.1, 66.7, 18.1, −1.2 (3C); HRMS (ES+, *m*/*z*): found 359.1349, calcd. for C_19_H_23_ClN_2_OSi, [M+H]^+^, 359.1346.

#### 3.2.2. 4-Chloro-2-(4-methoxyphenyl)-1-((2-(trimethylsilyl)ethoxy)methyl)-1*H*-pyrrolo[2,3-*b*]pyridine (**14b**)

The compound was made as described for **14a**, starting with 4-chloro-2-iodo-1-((2-(trimethylsilyl)ethoxy)methyl)-1*H*-pyrrolo[2,3-*b*]pyridine (**7**) (1.00 g, 2.44 mmol) and 4-methoxyphenylboronic acid (440 mg, 2.93 mmol). The reaction time was 30 min at 100 °C. The product was purified by silica-gel column chromatography (*n*-hexane/EtOAc, 4:1, R_f_ = 0.40) to give a yellow oil, 650 mg (1.67 mmol, 68%); ^1^H NMR (400 MHz, CDCl_3_) δ 8.18 (d, *J* = 5.2 Hz, 1H), 7.72 (d, *J* = 8.8 Hz, 2H), 7.11 (d, *J* = 5.2 Hz, 1H), 7.00 (d, *J* = 8.8 Hz, 2H), 6.61 (s, 1H), 5.63 (s, 2H), 3.84 (s, 3H), 3.74 (t, *J* = 8.3 Hz, 2H), 0.97 (t, *J* = 8.3 Hz, 2H), −0.04 (s, 9H); ^13^C NMR (100 MHz, CDCl_3_) δ 160.1, 150.2, 142.8 (2C), 134.9, 130.7 (2C), 123.9, 119.9, 116.9, 114.2 (2C), 98.2, 70.9, 66.5, 55.3, 18.0, −1.4 (3C); IR (neat, cm^−1^): 2951 (w), 2837 (w), 1613 (w), 1498 (m), 1246 (s), 1075 (m), 832 (s), 695 (w); HRMS (APCI/ASAP, *m*/*z*): found 389.1445, calcd. for C_20_H_26_N_2_O_2_SiCl, [M+H]^+^, 389.1452.

#### 3.2.3. (4-(4-Chloro-1-((2-(trimethylsilyl)ethoxy)methyl)-1*H*-pyrrolo[2,3-*b*]pyridin-2-yl)phenyl)methanol (**14c**)

The synthesis was performed as described for **14a**, starting with compound **7** (1.23 g, 3.02 mmol); (4-(hydroxymethyl)phenyl)boronic acid (539 mg, 3.55 mmol), but using Pd(PPh_3_)_4_ (180 mg, 0.156 mmol). The reaction was run at 80 °C for 9 h. The product was purified by silica-gel column chromatography (*n*-pentane/EtOAc, 4:1, R*_f_* = 0.27) giving a light-yellow oil, 936 mg (2.50 mmol, 79%); ^1^H NMR (400 MHz, DMSO-*d*_6_) δ 8.27 (d, *J* = 5.2 Hz, 1H), 7.77 (d, *J* = 8.3 Hz, 2H), 7.47 (d, *J* = 8.3 Hz, 2H), 7.33 (d, *J* = 5.2 Hz, 1H), 6.75 (s, 1H), 5.63 (s, 2H), 5.31 (t, *J* = 5.7 Hz, 1H), 4.58 (d, *J* = 5.7 Hz, 2H), 3.61 (t, *J* = 8.0 Hz, 2H), 0.84 (t, *J* = 8.0 Hz, 2H), −0.11 (s, 9H); ^13^C NMR (100 MHz, DMSO-*d*_6_) δ 149.8, 143.6, 143.4, 142.5, 133.8, 129.2, 128.7 (2C), 126.7 (2C), 118.9, 116.9, 98.2, 70.8, 65.9, 62.5, 17.3, −1.4 (3C); IR (neat, cm^−1^): 3330 (w, br), 2950 (m), 2893 (m), 1558 (m), 1368 (m), 1286 (s), 1120 (s), 930 (s, br), 909 (m, br), 831 (s, br)); HRMS (APCI/ASAP, *m*/*z*): found 389.1445, calcd. for C_20_H_26_N_2_O_2_SiCl, [M+H]^+^, 389.1452.

#### 3.2.4. 2-(4-(((*tert*-Butyldimethylsilyl)oxy)methyl)phenyl)-4-chloro-1-((2-(trimethylsilyl)ethoxy)methyl)-1*H*-pyrrolo[2,3-*b*]pyridine (**14d**)

The synthesis was performed as described as for **14a**, but starting with compound **7** (901 mg, 2.2 mmol), (4-(((*tert*-butyldimethylsilyl)oxy)methyl)phenyl)boronic acid (644 mg, 2.42 mmol), Pd(PPh_3_)_4_ (127 mg, 0.110 mmol). The mixture was stirred at 85 °C for 2 h. The product was purified by silica-gel column chromatography (*n*-pentane/EtOAc, 97:3, R*_f_* = 0.11) to give 991 mg (1.97 mmol, 90%) of a white solid; mp. 48.5–50 °C; ^1^H NMR (400 MHz, DMSO-*d*_6_) δ 8.27 (d, *J* = 5.2 Hz, 1H), 7.79 (d, *J* = 8.3 Hz, 2H), 7.46 (d, *J* = 8.3 Hz, 2H), 7.33 (d, *J* = 5.2 Hz, 1H), 6.76 (s, 1H), 5.64 (s, 2H), 4.80 (s, 2H), 3.59 (t, *J* = 8.0 Hz, 2H), 0.93 (s, 9H), 0.82 (t, *J* = 8.0 Hz, 2H), 0.11 (s, 6H), −0.11 (s, 9H); ^13^C NMR (100 MHz, DMSO-*d*_6_) δ 149.8, 143.5, 142.4, 142.2, 133.9, 129.4, 128.8 (2C), 126.3 (2C), 118.9, 117.0, 98.3, 70.8, 65.9, 63.9, 25.8 (3C), 18.0, 17.3, −1.5 (3C), −5.3 (2C); IR (neat, cm^−1^): 2951 (m), 2926 (m), 2883 (w), 2853 (w), 1554 (m), 1471 (m), 1458 (m), 1375 (m), 1249 (s), 1074 (s), 918 (m), 829 (s, br), 767 (s, br); HRMS (APCI/ASAP, *m*/*z*): found 503.2315, calcd. for C_26_H_40_N_2_O_2_Si_2_Cl, [M+H]^+^, 503.2317.

#### 3.2.5. 4-Chloro-2-(4-(((2-(trimethylsilyl)ethoxy)methoxy)methyl)phenyl)-1-((2-(trimethylsilyl)ethoxy)methyl)-1*H*-pyrrolo[2,3-*b*]pyridine (**14e**)

(4-(4-Chloro-1-((2-(trimethylsilyl)ethoxy)methyl)-1*H*-pyrrolo[2,3-*b*]pyridin-2-yl)phenyl)-methanol (**14c**) (1.05 g, 2.70 mmol) was dissolved in dry DMF (11 mL). NaH (114 mg, 4.75 mmol) was added under an N_2_ atmosphere at 0 °C. After stirring for 30 min, 2-(trimethylsilyl)ethoxymethyl chloride (0.7 mL, 3.96 mmol) was added dropwise over 10 min. The mixture was further stirred at 0 °C for 4 h and 45 min, before being allowed to warm to 22 °C. The mixture was quenched with sat. aq. NH_4_Cl (100 mL) and extracted with EtOAc (3 × 100 mL). The combined organic phases were washed with brine (100 mL), dried over anhydrous Na_2_SO_4_, filtered and concentrated in vacuo. The product was purified by silica-gel column chromatography (*n*-pentane/EtOAc, 96:4, R*_f_* = 0.19) giving 717 mg (1.38 mmol, 51%) of a clear oil. ^1^H NMR (400 MHz, DMSO-*d*_6_) δ: 8.28 (d, *J* = 5.2 Hz, 1H), 7.80 (d, *J* = 8.2 Hz, 2H), 7.48 (d, *J* = 8.2 Hz, 2H), 7.34 (d, *J* = 5.2 Hz, 1H), 6.77 (s, 1H), 5.65 (s, 2H), 4.73 (s, 2H), 4.62 (s, 2H), 3.62 (t, *J* = 8.2 Hz, 2H), 3.60 (t, *J* = 8.0 Hz, 2H), 0.90 (t, *J* = 8.2 Hz, 2H), 0.83 (t, *J* = 8.0 Hz, 2H), 0.01 (s, 9H), −0.11 (s, 9H); ^13^C NMR (100 MHz, DMSO-*d*_6_) δ 149.8, 143.6, 142.2, 139.3, 133.9, 129.9, 128.9 (2C), 127.9 (2C), 118.8, 117.0, 98.5, 93.9, 70.8, 68.2, 65.9, 64.4, 17.6, 17.3, −1.3 (3C), −1.5 (3C); IR (neat, cm^−1^): 2951 (w), 2892 (w, br), 1557 (w), 1369 (m), 1248 (s), 1157 (m), 1055 (s, br), 1023 (s, br), 910 (m), 855 (s), 830 (s), 756 (m); HRMS (APCI/ASAP, *m*/*z*): found 519.2262, calcd. for C_26_H_40_N_2_O_3_Si_2_Cl, [M+H]^+^, 519.2266.

#### 3.2.6. N-Benzyl-*N*-methyl-2-phenyl-1-((2-(trimethylsilyl)ethoxy)methyl)-1*H*-pyrrolo[2,3-*b*]pyridin-4-amine (**13a**) via **12**

A mixture of *N*-benzyl-2-iodo-*N*-methyl-1-((2-(trimethylsilyl)ethoxy)methyl)-1*H*-pyrrolo[2,3-*b*]-pyridin-4-amine (**12**) (49 mg, 0.10 mmol), phenylboronic acid (16 mg, 0.13 mmol), XPhos (2.9 mg, 6.1 µmol), XPhos Pd G2 (4.1 mg, 5.2 µmol), K_2_CO_3_ (42 mg, 0.31 mmol) was added to degassed 1,4-dioxane (1 mL) and water (0.5 mL) under an N_2_ atmosphere. The mixture was stirred at 100 °C for 1 h before being allowed to cool to 22 °C. The solvent was removed in vacuo before EtOAc (20 mL) and water (15 mL) were added to the mixture, and the layers were separated. The aqueous layer was extracted with EtOAc (2 × 20 mL). The combined organic phases were washed with brine (20 mL), dried over anhydrous Na_2_SO_4_, filtered and concentrated in vacuo. The product was purified by silica-gel column chromatography (*n*-pentane/EtOAc, 9:1, R_f_ = 0.21) to give 34 mg (0.077 mmol, 77%) of a yellow oil.^1^H NMR (400 MHz, CDCl_3_) δ 8.07 (d, *J* = 5.6 Hz, 1H), 7.67 (ap.d, 2H), 7.43–7.39 (m, 2H), 7.36–7.24 (m, 6H), 6.58 (s, 1H), 6.34 (d, *J* = 5.6 Hz, 1H), 5.63 (s, 2H), 4.81 (s, 2H), 3.72 (t, *J* = 8.3 Hz, 2H), 3.19 (s, 3H), 0.95 (t, *J* = 8.3 Hz, 2H), −0.04 (s, 9H); ^13^C NMR (100 MHz, CDCl_3_) δ 151.4, 150.4, 144.4, 138.1, 137.8, 132.5, 129.1 (2C), 128.7 (2C), 128.5 (2C), 127.8, 127.2, 126.8 (2C), 108.4, 101.2, 100.8, 70.8, 66.1, 57.2, 39.2, 18.1, −1.4 (3C); IR (neat, cm^−1^): 3061 (w), 2950 (w), 2893 (w, br), 1577 (s), 1347 (w), 1071 (m), 1027 (w), 835 (m), 697 (m); HRMS (APCI/ASAP, *m*/*z*): found 444.2462, calcd. for C_27_H_34_N_3_OSi, [M+H]^+^, 444.2471.

#### 3.2.7. *N*-Benzyl-*N*-methyl-2-phenyl-1-((2-(trimethylsilyl)ethoxy)methyl)-1*H*-pyrrolo[2,3-*b*]pyridin-4-amine (**13a**) via **14a**

A mixture of 4-chloro-2-phenyl-1-((2-(trimethylsilyl)ethoxy)methyl)-1*H*-pyrrolo[2,3-*b*]-pyridine (**14a**) (230 mg, 0.640 mmol), *N*-benzylmethylamine (0.085 mL, 0.704 mmol), NaO*t*-Bu (185 mg, 1.92 mmol), RuPhos (15 mg, 0.032 mmol), Pd(OAc)_2_ (7.2 mg, 0.032 mmol) and *t*-BuOH (3 mL) were added under an N_2_ atmosphere. The reaction mixture was stirred at 85 °C for 1 h before being allowed to cool to rt. The solvent was removed in vacuo, and the mixture was added to EtOAc (15 mL) and water (15 mL). The layers were separated, and the aqueous layer was extracted with more EtOAc (3 × 15 mL). The combined organic phases were washed with brine (20 mL), dried over anhydrous Na_2_SO_4_, filtered and concentrated in vacuo. The product was purified by silica-gel column chromatography (*n*-hexane/EtOAc, 7:3, R_f_ = 0.20) to give 210 mg (0.473 mmol, 74%) of a yellow oil. The ^1^H NMR corresponded with that reported above.

#### 3.2.8. *N*-Benzyl-2-(4-methoxyphenyl)-*N*-methyl-1-((2-(trimethylsilyl)ethoxy)methyl)-1*H*-pyrrolo[2,3-*b*]-pyridin-4-amine (**13b**)

The synthesis was done as described for **13a** starting with **14b**. The reaction time was 1 h at 85 °C. Purification by silica-gel column chromatography (*n*-hexane/EtOAc, 7:3, *R_f_* = 0.41) gave 600 mg (1.26 mmol, 76%) of a yellow oil, ^1^H NMR (400 MHz, CDCl_3_) δ 8.05 (d, *J* = 5.6 Hz, 1H), 7.61 (d, *J* = 8.8 Hz, 2H), 7.36–7.24 (m, 5H), 6.95 (d, *J* = 8.8 Hz, 2H), 6.51 (s, 1H), 6.33 (d, *J* = 5.6 Hz, 1H), 5.61 (s, 2H), 4.80 (s, 2H), 3.83 (s, 3H), 3.74 (t, *J* = 8.3 Hz, 2H), 3.18 (s, 3H), 0.96 (t, *J* = 8.3 Hz, 2H), −0.03 (s, 9H); ^13^C NMR (100 MHz, CDCl_3_) δ 159.5, 151.2, 150.3, 144.0, 138.1, 137.7, 130.5 (2C), 128.7 (2C), 127.2, 126.8 (2C), 124.9, 114.0 (2C), 108.5, 100.8, 100.4, 70.8, 66.1, 57.1, 55.3, 39.1, 18.1, −1.4 (3C); IR (neat, cm^−1^): 2950 (w), 2836 (w, br), 1576 (m), 1495 (m), 1345 (w), 1069 (m), 832 (s); HRMS (APCI/ASAP, *m*/*z*): found 474.2570, calcd. for C_28_H_36_N_3_O_2_Si, [M+H]^+^, 474.2577.

#### 3.2.9. (4-(4-(Benzyl(methyl)amino)-1-((2-(trimethylsilyl)ethoxy)methyl)-1*H*-pyrrolo[2,3-*b*]pyridin-2-yl)phenyl)methanol (**13c**)

Compound **13c** was isolated as a minor product of the reaction described in preparation of **13d** starting with **14d**. Purification by silica-gel column chromatography (*n*-pentane/EtOAc, 9:1, R*_f_* = 0.01) gave 45 mg (0.095 mmol, 6%) of a brownish wax. ^1^H NMR (400 MHz, DMSO-*d*_6_) δ 7.91 (d, *J* = 5.7 Hz, 1H), 7.64 (d, *J* = 8.2 Hz, 2H), 7.39 (d, *J* = 8.2 Hz, 2H), 7.35–7.31 (m, 2H), 7.27–7.24 (m, 3H), 6.68 (s, 1H), 6.34 (d, *J* = 5.7 Hz, 1H), 5.55 (s, 2H), 5.24 (t, *J* = 5.7 Hz, 1H), 4.85 (s, 2H), 4.54 (d, *J* = 5.7 Hz, 2H), 3.63 (t, *J* = 8.2 Hz, 2H), 3.25 (s, 3H), 0.85 (t, *J* = 8.2 Hz, 2H), −0.08 (s, 9H); ^13^C NMR (100 MHz, DMSO-*d*_6_) δ 151.1, 149.4, 143.9, 142.3, 138.4, 136.6, 130.3, 128.6 (2C), 128.2 (2C), 126.9, 126.6 (2C), 126.5 (2C), 107.4, 101.3, 100.5, 70.4, 65.5, 62.6, 56.1, 39.8, 17.4, −1.4 (3C); IR (neat, cm^−1^): 3400 (w, br), 2927 (w), 2853 (w), 1737 (w), 1578 (s), 1496 (m), 1354 (m), 1245 (s), 1205 (m), 1064 (s, br), 832 (s, sh), 695 (s); HRMS (APCI/ASAP, *m*/*z*): found 474.2571, calcd. for C_28_H_36_N_3_O_2_Si, [M+H]^+^, 474.2577.

#### 3.2.10. *N*-Benzyl-2-(4-(((*tert*-butyldimethylsilyl)oxy)methyl)phenyl)-*N*-methyl-1-((2-(trimethylsilyl)ethoxy)methyl)-1*H*-pyrrolo[2,3-*b*]pyridin-4-amine (**13d**)

A mixture of 2-(4-(((*tert*-butyldimethylsilyl)oxy)methyl)phenyl)-4-chloro-1-((2-(trimethylsilyl)ethoxy) methyl)1*H*-pyrrolo[2,3-*b*]pyridine (**14d**) (816 mg, 1.62 mmol), *N*-methyl-1-phenylmethylamine (1.5 mL, 11.6 mmol), NaO*t*-Bu (470 mg, 4.89 mmol), RuPhos (51 mg, 0.109 mmol) and Pd(OAc)_2_ (24 mg, 0.107 mmol) was added to degassed *t*-BuOH (28 mL), under an N_2_ atmosphere. The reaction mixture was stirred at 85 °C for 30 min before being cooled to rt. The pH was adjusted to 6 with (NH_4_)_2_SO_4_ (30% in H_2_O). The solvent was removed in vacuo. CH_2_Cl_2_ (50 mL) and water (50 mL) were added to the flask and the layers were separated. The water phase was extracted with more CH_2_Cl_2_ (3 × 50 mL). The combined organic phases were washed with brine (50 mL), dried over anhydrous Na_2_SO_4_, filtered and concentrated in vacuo. The product was purified by silica-gel column chromatography (*n*-pentane/EtOAc, 9:1, R*_f_* = 0.27). This gave 847 mg (1.44 mmol, 89%) of a light-yellow oil. ^1^H NMR (400 MHz, DMSO-*d*_6_) δ 7.91 (d, *J* = 5.8 Hz, 1H), 7.65 (d, *J* = 8.3 Hz, 2H), 7.39–35 (m, 2H), 7.33–31 (m, 2H), 7.27–7.24 (m, 3H), 6.69 (s, 1H), 6.33 (d, *J* = 5.8 Hz, 1H), 5.55 (s, 2H), 4.85 (s, 2H), 4.75 (s, 2H), 3.60 (t, *J* = 8.1 Hz, 2H), 3.25 (s, 3H), 0.92 (s, 9H), 0.83 (t, *J* = 8.0 Hz, 2H), 0.09 (s, 6H), −0.09 (s, 9H); ^13^C NMR (100 MHz, DMSO-d6) δ 151.1, 149.4, 144.0, 140.9, 138.4, 136.4, 130.6, 128.6 (2C), 128.3 (2C), 126.9, 126.6 (2C), 126.1 (2C), 107.4, 101.5, 100.5, 70.4, 65.5, 63.9, 56.1, 39.9 25.8 (3C), 18.0, 17.4, −1.4 (3C), −5.3 (2C); IR (neat, cm^−1^): 2951 (m), 2928 (m), 2887 (w), 2855 (w), 1702 (w), 1577 (m), 1496 (m), 1248 (m), 1072 (s, br), 831 (s, br), 774 (s, br), 694 HRMS (APCI/ASAP, *m*/*z*): found 588.3446, calcd. for C_34_H_50_N_3_O_2_S_i2_, [M+H]^+^, 588.3442.

#### 3.2.11. *N*-Benzyl-*N*-methyl-2-(4-(((2-(trimethylsilyl)ethoxy)methoxy)methyl)phenyl)-1-((2-(trimethylsilyl)ethoxy)methyl)-1*H*-pyrrolo[2,3-*b*]pyridin-4-amine (**13e**)

A mixture of 4-chloro-2-(4-(((2-(trimethylsilyl) ethoxy)methoxy)methyl)phenyl)-1-((2-(trimethyl-silyl)ethoxy)methyl)-1*H*-pyrrolo[2,3-*b*]pyridine (**14e**) (111 mg, 0.213 mmol), *N*-methyl-1-phenylmethylamine (0.2 mL, 1.55 mmol), NaO*t*-Bu (71 mg, 0.739 mmol), RuPhos (6 mg, 0.013 mmol) and Pd(OAc)_2_ (4 mg, 0.018 mmol) was added to degassed *t*-BuOH (3 mL), under an N_2_ atmosphere. The reaction mixture was stirred at 85 °C for 5 h before being cooled to room temperature. The solvent was removed in vacuo. CH_2_Cl_2_ (15 mL) and water (15 mL) were added to the flask and the layers were separated. The water phase was then adjusted to pH 7 with *sat. aq*. NH_4_Cl and extracted with CH_2_Cl_2_ (3 × 15 mL). The combined organic phases were washed with brine (15 mL), dried over anhydrous Na_2_SO_4_, filtered and concentrated in vacuo. The product was purified by silica-gel column chromatography (*n*-pentane/EtOAc, 9:1, R*_f_* = 0.19) to give 118 mg (0.196 mmol, 92%) of an oil. ^1^H NMR (400 MHz, DMSO-*d*_6_) δ 7.91 (d, *J* = 5.7 Hz, 1H), 7.66 (d, *J* = 8.2 Hz, 2H), 7.39 (d, *J* = 8.2 Hz, 2H), 7.35–7.31 (m, 2H), 7.27–7.24 (m, 3H), 6.70 (s, 1H), 6.35 (d, *J* = 5.8 Hz, 1H), 5.55 (s, 2H), 4.86 (s, 2H), 4.70 (s, 2H), 4.57 (s, 2H), 3.62 (t, *J* = 3.9 Hz, 2H), 3.59 (t, *J* = 4.4 Hz, 2H), 3.25 (s, 3H), 0.88 (t, *J* = 8.2 Hz, 2H), 0.84 (t, *J* = 7.9 Hz, 2H), 0.00 (s, 9H), −0.09 (s, 9H); ^13^C NMR (100 MHz, DMSO-*d*_6_) δ 151.1, 149.4, 144.1, 138.4, 137.9, 136.3, 131.1, 128.6 (2C), 128.3 (2C), 127.8 (2C), 126.9, 126.6 (2C), 107.3, 101.6, 100.5, 93.8, 70.4, 68.2, 65.5, 64.4, 56.1, 39.9, 17.5, 17.4, −1.3 (3C), −1.4 (3C); IR (neat, cm^−1^, neat) 2950 (w), 2883 (w, br), 1576 (s), 1495 (m), 1452 (m), 1245 (m), 1100 (s, br), 832 (s, br); HRMS (APCI/ASAP, *m*/*z*): found 604.3390, calcd. for C_34_H_50_N_3_O_3_Si_2_, [M+H]^+^, 604.3391.

#### 3.2.12. *N*-Benzyl-*N*-methyl-2-phenyl-1*H*-pyrrolo-[2,3-*b*]pyridin-4-amine (**3a**)

Compound **13a** (140 mg, 0.316 mmol) was dissolved in dry CH_2_Cl_2_ (10 mL), and TFA (3 mL) was added dropwise over 5 min under N_2_ atmosphere. The reaction mixture was stirred at rt for 9.5 h. The solvent was removed in vacuo and the resulted crude was dissolved in THF (5 mL) and a saturated NaHCO_3_ solution (5 mL) was added dropwise over 10 min, and the mixture was stirred at rt for 18 h. Upon completion, the solvent was removed under reduced pressure and extracted with EtOAc (3 × 15 mL). The combined organic phases were washed with brine (10 mL), dried over anhydrous Na_2_SO_4_, filtered and concentrated in vacuo. The product was purified by silica-gel column chromatography (EtOAc, R*_f_* = 0.30) giving 40 mg (0.128 mmol, 41%) of a pale-yellow solid; *mp.* 191–193 °C. ^1^H NMR (600 MHz, DMSO-*d*_6_) *δ*: 12.23 (s, 1H), 7.87 (d, *J* = 6.1 Hz, 1H), 7.83–7.81 (m, 2H), 7.43–7.40 (m, 2H), 7.37–7.34 (m, 2H), 7.30–7.25 (m, 2H), 7.08 (s, 1H), 6.35 (d, *J* = 6.2 Hz, 1H), 4.93 (s, 2H), 3.36 (s, 3H); ^13^C NMR (151 MHz, DMSO-*d*_6_) *δ* 150.5, 140.8, 137.8, 133.8, 131.4, 128.8 (2C), 128.6 (2C), 127.4, 127.0 (2C), 126.6 (2C), 124.8 (2C), 108.5, 99.6, 99.1, 56.2, 40.4; IR (neat, cm^−1^): 2923 (m), 1596 (s), 1453 (m), 1199 (w), 932 (m); HRMS (ES+, *m*/*z*): found 314.1661, calcd. for C_21_H_19_N_3_, [M+H]^+^, 314.1657.

#### 3.2.13. *N*-Benzyl-*N*-methyl-2-(4-methoxyphenyl-1*H*-pyrrolo-[2,3-*b*]pyridin-4-amine (**3b**)

*N*-benzyl-2-(4-methoxyphenyl)-*N*-methyl-1-((2-(trimethylsilyl)ethoxy)methyl)-1*H*-pyrrolo-[2,3-*b*]-pyridin-4-amine (**16**) (94 mg, 0.20 mmol) was dissolved in CH_2_Cl_2_ (6.6 mL) and added to TFA (1.3 mL, 17 mmol) under an N_2_ atmosphere. The mixture was stirred at 22 °C for 6 h before allowed to cool to rt. The solvent was removed in vacuo before THF (10 mL) and sat. aq. NaHCO_3_ (10 mL) was added to the residue. The mixture was stirred at 22 °C for 16 h. As analysis indicated incomplete conversion, more sat. aq. NaHCO_3_ (5 mL) was added, and the reaction was stirred for an additional 6 h, before the solvent was removed in vacuo. The residue was partitioned between sat. aq. NH_4_Cl (15 mL) and EtOAc (15 mL), and the layers were separated. The aqueous layer was extracted with more EtOAc (2 × 15 mL), and the combined organic phases were washed with brine (15 mL), dried over anhydrous Na_2_SO_4_, filtered and concentrated in vacuo. The product was purified twice by silica-gel column chromatography (CH_2_Cl_2_/MeOH, 97:3, R_f_ = 0.07) to give 9 mg (0.026 mmol, 14%) of a yellow oil being semi-pure. ^1^H NMR (400 MHz, CDCl_3_) δ 12.16 (s, br, 1H), 7.95 (d, *J* = 5.8 Hz, 1H), 7.63 (d, *J* = 8.7 Hz, 2H), 7.38–7.30 (m, 5H), 6.92 (d, *J* = 8.7 Hz, 2H), 6.63 (s, 1H), 6.26 (d, *J* = 5.8 Hz, 1H), 4.84 (s, 2H), 3.83 (s, 3H), 3.24 (s, 3H); ^13^C NMR (100 MHz, CDCl_3_) δ 159.2, 150.9 (2C), 142.9, 137.8, 135.0, 128.8 (2C), 127.3, 126.8 (2C), 126.6 (2C), 125.0, 114.4 (2C), 109.8, 99.9, 96.9, 57.1, 55.4, 39.3; IR (neat, cm^−1^): 3133 (w, br), 2926 (w), 2852 (w, br), 1600 (m), 1500 (s), 1249 (m), 833 (w), 697 (w); HRMS (APCI/ASAP, *m*/*z*): found 343.1685, calcd. for C_22_H_21_N_3_O, [M+H]^+^, 343.1685.

#### 3.2.14. (4-(4-(Benzyl(methyl)amino)-1*H*-pyrrolo[2,3-*b*]pyridin-2-yl)phenyl)methanol (**3c**)

*N*-Benzyl-2-(4-(((*tert*-butyldimethylsilyl)oxy)methyl)phenyl)-*N*-methyl-1-((2-(trimethylsilyl)ethoxy)methyl)-1*H*-pyrrolo[2,3-*b*]pyridin-4-amine (**13d**) (430 mg, 0.731 mmol) was dissolved in dry CH_2_Cl_2_ (30 mL) and TFA (5 mL, 65.3 mmol) was added dropwise over a period of 5 min under an N_2_ atmosphere. The reaction mixture was stirred at room temperature for 4.5 h. The solvent was removed in vacuo. The mixture was dissolved in THF (30 mL) and NaHCO_3_ (*sat. aq.*, 40 mL) was added dropwise over 10 min. The mixture was then stirred at room temperature for 19.5 h, before the solvent was removed in vacuo. The product was purified by silica-gel column chromatography (CH_2_Cl_2_/MeOH, 9:1, R*_f_* = 0.15) to give an off-white powder, 186 mg (0.542 mmol, 74%). This material was dissolved in a mixture of MeOH and CHCl_3_ (1:1 by vol, 2 mL). Upon storage at 4 °C for 18 h, crystals formed. Slow evaporation of solvent gave after drying 128 mg (0.373 mmol 51%) of an off-white solid, mp. 200–201 °C; ^1^H NMR (600 MHz, DMSO-d*_6_*) δ 11.84 (s, 1H), 7.83 (d, *J* = 5.8 Hz, 1H), 7.77 (d, *J* = 8.2 Hz, 2H), 7.36–7.32 (m, 4H), 7.28–7.25 (m, 3H), 6.97 (s, 1H), 6.24 (d, *J* = 5.8 Hz, 1H), 5.17 (t, *J* = 5.7 Hz, 1H), 4.87 (s, 2H), 4.50 (d, *J* = 5.7 Hz, 2H), 3.27 (s, 3H); ^13^C NMR (150 MHz, DMSO-d6) δ 150.5, 149.6, 143.0, 141.5, 138.4, 133.5, 130.3, 128.6 (2C), 126.9, 126.8 (2C), 126.7 (2C), 124.4 (2C), 108.6, 99.5, 98.0, 62.6, 56.1, 40.1; IR (neat, cm^−1^): 3204 (m, br), 3063 (m, br), 3023 (m, br), 2919 (m), 2852 (m), 1574 (s), 1518 (s), 1499 (m), 1371 (m), 1294 (m), 1199 (s), 1027 (s), 1013 (s), 1013 (s, sh), 767 (s, br), 723 (s, sh); HRMS (APCI/ASAP, *m*/*z*): found 344.1757, calcd. for C_22_H_22_N_3_O, [M+H]^+^, 344.1763.

#### 3.2.15. 6-Methyl-1-phenyl-2,6,7,12-tetrahydro-2,3,6-triazabenzo[6,7]cycloocta[1,2,3-cd]indene (**16a**)

Following the reaction to obtain compound **3a**, purification with silica-gel column chromatography (EtOAc, R_f_ = 0.10) gave 12 mg (0.039 mmol, 12%) as a light-yellow solid, mp. 291–293 °C; ^1^H NMR (600 MHz, DMSO-*d*_6_) *δ* 11.51 (s, 1H), 7.80 (d, *J* = 5.7 Hz, 1H), 7.62–7.55 (m, 5H), 7.46–7.43 (m, 1H), 7.25–7.22 (m, 1H), 7.15–7.13 (m, 1H), 7.02 (dd, *J* = 7.6 Hz, 1.3 Hz, 1H), 6.16 (d, *J* = 5.8 Hz, 1H), 4.70 (s, 2H), 4.15 (s, 2H), 3.34 (s, 3H); ^13^C NMR (151 MHz, DMSO-*d*_6_) *δ* 149.3, 140.3, 137.1, 132.8 (2C), 131.4 (2C), 129.8 (2C), 128.5 (3C), 128.4, 127.7, 127.4, 127.0, 110.7, 106.3, 98.9, 53.9, 40.8, 31.6; HRMS (ES+, *m*/*z*): found 326.1661, calcd for C_22_H_19_N_3_, [M+H]^+^, 326.1657.

#### 3.2.16. 1-(4-Methoxyphenyl)-6-methyl-2,6,7,12-tetrahydro-2,3,6-triazabenzo[6,7]cycloocta[1,2,3-cd]indene (**16b**)

Compound **13b** (100 mg, 0.211 mmol) was dissolved in dry CH_2_Cl_2_ (5 mL) and TFA (1 mL) was added dropwise over 5 min under an N_2_ atmosphere. The reaction mixture was stirred at 50 °C for 18 h. The solvent was removed in vacuo. The mixture was dissolved in THF (5 mL) and saturated NaHCO_3_ solution (5 mL) was added dropwise over 10 min. The mixture was then stirred at rt for 4 h. The solvent was removed in vacuo, and water (15 mL) was added to the mixture followed by extraction with EtOAc (3 × 25 mL). The combined organic phases were washed with brine (10 mL), dried over anhydrous Na_2_SO_4_, filtered and concentrated in vacuo. The product was purified by silica-gel column chromatography (EtOAc/*n*-hexane, 9.5:0.5, R*_f_* = 0.30), giving 52 mg (0.146 mmol, 69%) of an off-white solid; mp 287–289 °C (decomp.); ^1^H NMR (600 MHz, DMSO-*d*_6_) *δ* 11.38 (s, 1H), 7.75 (d, *J* = 5.7 Hz, 1H), 7.55 (d, *J* = 7.5 Hz, 1H), 7.50 (d, *J* = 8.1 Hz, 2H), 7.20 (t, *J* = 7.5 Hz, 1H), 7.10 (d, *J* = 8.3 Hz, 3H), 6.98 (d, *J* = 7.6 Hz, 1H), 6.11 (d, *J* = 5.8 Hz, 1H), 4.66 (s, 2H), 4.10 (s, 2H), 3.82 (s, 3H), 3.30 (s, 3H); ^13^C NMR (151 MHz, DMSO-*d*_6_) *δ* 158.6, 150.2, 149.0, 143.0, 140.4, 137.1, 131.3, 130.9 (2C), 128.5, 128.3, 127.7, 126.9, 125.1, 113.9 (2C), 109.9, 106.3, 98.8, 55.2, 53.9, 40.8, 31.7. HRMS (ES+, *m*/*z*): found 356.1763, calcd for C_23_H_21_N_3_O [M+H]^+^, 356.1763.

#### 3.2.17. (4-(6-Methyl-2,6,7,12-tetrahydro-2,3,6-triazabenzo[6,7]cycloocta[1,2,3-cd]inden-1-yl)phenyl)methanol (**16c**)

Compound **13e** (182 mg, 0.301 mmol) was dissolved in dry CH_2_Cl_2_ (40 mL) and added to TFA (2 mL, 26.2 mmol) under an N_2_ atmosphere. The reaction mixture was stirred at 50 °C for 2 h, before it was cooled to rt, and the solvent was removed in vacuo. The mixture was dissolved in THF (20 mL), before NaHCO_3_ (*sat. aq.*, 20 mL) was added dropwise over 10 min. The mixture was then stirred at 22 °C for 22 h. The solvent was removed in vacuo, and the mixture was dissolved in CH_2_Cl_2_ (40 mL) and MeOH (20 mL) and stirred at rt for 1.5 h. The reaction mixture was then filtered, and solvent was removed in vacuo. The mixture was dissolved in MeOH (10 mL) before NH_3_ (12.5% in water, 20 mL, 133.6 mmol) was added dropwise over a period of 10 min. The mixture was stirred at rt for 22 h. The solvent was removed in vacuo, giving a yellow powder. The mixture was purified using silica-gel column chromatography (CH_2_Cl_2_, 9:1). This gave 12 mg (0.034 mmol, 11%) of **17c** (R*_f_* = 0.08) as a white solid. ^1^H NMR (600 MHz, DMSO-*d*_6_) δ 11.65 (s, 1H), 7.81 (m, 1H), 7.60–7.56 (m, 3H), 7.50 (d, *J* = 8.1 Hz, 2H), 7.24 (t, *J* = 7.4 Hz, 1H), 7.14 (t, *J* = 7.3 Hz, 1H), 7.00 (d, *J* = 7.3 Hz, 1H), 6.20 (d, *J* = 5.8 Hz, 1H), 5.30 (m, 1H), 4.71 (s, 2H), 4.61 (d, *J* = 4.2 Hz, 2H), 4.16 (s, 2H), 3.25 (s, 3H); ^13^C NMR (100 MHz, DMSO-*d*_6_) δ 150.3, 149.7, 142.1, 142.0, 140.4, 136.9, 131.6, 131.0, 129.5 (2C), 128.6, 128.4, 127.9, 127.1, 126.6 (2C), 110.8, 106.4, 99.0, 62.7, 48.6, 41.0, 31.7; IR (cm^−1^, neat): 3606 (w), 3460 (w, br), 3102 (w), 3024 (w), 3000 (w), 2947 (w), 2842 (m), 1596 (s), 1557 (s), 1540 (s), 1518 (s), 1460 (m), 1376 (m), 1342 (m), 1206 (m), 1097 (m), 1043 (s, br), 1015 (m), 917 (s, sh), 783 (s, sh), 756 (s, sh); HRMS (APCI/ASAP, *m*/*z*): found 356.1759, calcd. for C_23_H_22_N_3_O, [M+H]^+^, 356.1763. The material contains residual CH_2_Cl_2_ from purification. Also, isolated was 13 mg (0.038 mmol, 13%) of **3c** (R*_f_* = 0.21) as a white powder; mp. 199–201 °C. The spectral data were identical to that reported above.

#### 3.2.18. 3,3-Methylenebis(*N*-benzyl-*N*-methyl-2-phenyl-1*H*-pyrrolo[2,3-*b*]pyridin-4-amine (**17a**)

Compound **3a** (40 mg, 0.128 mmol) in dry CH_2_Cl_2_ (0.5 mL) was added to formaldehyde (5 mg,0.153 mmol) and TFA (0.012 mL, 0.153 mmol) under inert condition. The reaction mixture was stirred at rt for 6.5 h. Then the solvent was removed under reduced pressure, the mixture was neutralized with saturated NaHCO_3_ solution (5 mL) and extracted with EtOAc (3 × 10 mL). The layers were separated, and the combined organic phases were washed with brine (10 mL), dried over anhydrous Na_2_SO_4_, filtered and concentrated in vacuo. Purification by silica-gel column chromatography gave 13 mg (0.020 mmol, 32%) of a solid; mp. 276–278 °C; ^1^H NMR (600 MHz, DMSO-*d*_6_) *δ* 11.04 (s, 2H), 7.86 (d, *J* = 5.8 Hz, 2H), 7.24–7.16 (m, 10H), 7.03 (d, *J* = 12 Hz, 4H), 6.91–6.88 (m, 2H), 6.50 (m, *J* = 5.3 Hz, 2H), 5.08 (s, 2H), 4.70 (s, 2H), 4.18 (s, 2H), 2.58 (s, 3H) ^13^C NMR (151 MHz, DMSO-*d*_6_) *δ* 153.6 (2C), 149.2 (2C), 142.7 (2C), 137.7 (2C), 133.4 (2C), 132.0 (2C), 128.2 (4C), 127.8 (4C),127.4 (4C), 126.8 (2C), 126.5 (4C), 126.1 (2C), 114.2 (2C), 110.3 (2C), 104.7 (2C), 59.1 (2C), 40.3 (2C), 23.7. HRMS (ES+, *m*/*z*): found 639.3234, calcd for C_43_H_38_N_6_, [M+H]+, 639.3236.

## 4. Conclusions

Two different routes were investigated for the preparation of the 7-azaindole **3c,** needed in SAR studies of CSF1R inhibition. The best strategy involved a regioselective Suzuki–Miyaura cross-coupling on C-2 catalysed by Pd(PPh_3_)_4_ and a Buchwald–Hartwig amination at C-4 employing RuPhos as a palladium ligand. The latter transformation only proceeds smoothly in the absence of acidic protons. Both inconveniently and interestingly the last SEM-deprotection proved most difficult, as the release of formaldehyde caused a side reaction. The major side product was a new type of 8-membered ring. By extending the reaction time in the TFA step, preparative useful reaction was seen for the methoxy derivative. Finally, CSF1R inhibition studies proved that the 7-azaindole **3c** was less active than the corresponding pyrrolopyrimidine.

## Data Availability

Data are available in the Appendix A.

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
