# Peer review of "Synthetic Routes to 2-aryl-1H-pyrrolo[2,3-b]pyridin-4-amines: Cross-Coupling and Challenges in SEM-Deprotection"

_molecules, 2024, doi:10.3390/molecules29194743_

Round 1

Reviewer 1 Report

Comments and Suggestions for Authors

The manuscript by Hoff and coauthors reports two methods to synthesize 2-aryl-1H-pyrrolo[2,3-b]pyridin-4-amines with moderate yields. In addition, CSF1R inhibition studies were also conducted, although the obtained 7-azaindole analogue 3c was less active than pyrrolopyrimidine and thienopyrimidine. A strength of this manuscript is testing a series of protocols for the preparation of 7-azaindoles, particularly the analogue 13c, by leveraging cross-couplings. Overall, this manuscript is a good research work, and suitable for publication in Molecules if the following issues are addressed.

1.The authors conducted the iodination of compound 10 with considerably low efficiency. What about freshly prepared LDA? Besides, n-butyllithium is a good substitution of LDA in this reaction. Further experiments are required.

2.The paragraph discussing Suzuki-Miyaura cross-coupling (page 4, lines 128-135)—“no reduction product (7)” and “reduced derivative 7 (entry 8)” should be corrected to “no reduction product (6)” and “reduced derivative 6 (entry 8)” respectively. Table 1 is suggested to make correction accordingly as well.

3.Figure 2 is fuzzy, a high resolution figure is required.

Author Response

Comment 1.The authors conducted the iodination of compound 10 with considerably low efficiency. What about freshly prepared LDA? Besides, n-butyllithium is a good substitution of LDA in this reaction. Further experiments are required.

Response 1: We trust the quality of LDA, as other sucessful iodinations were perfomed with the same batch. The use of n-BuLi is a viable option. However, we do not have available starting material to test this option within a reasonable time frame. To adresse this issue we have added the following text: The use of n-butyl lithium could lead to better results. This was however not tested.

Comment 2.The paragraph discussing Suzuki-Miyaura cross-coupling (page 4, lines 128-135)—“no reduction product (7)” and “reduced derivative 7 (entry 8)” should be corrected to “no reduction product (6)” and “reduced derivative 6 (entry 8)” respectively. Table 1 is suggested to make correction accordingly as well.

Response 2: This has been corrected.

Comment 3.Figure 2 is fuzzy, a high resolution figure is required.

Response 3: Larger fonts have been used in Figure 2.

Reviewer 2 Report

Comments and Suggestions for Authors

The present manuscript describes the pathway towards the preparation of 7-azaindole derivatives as the structural analogues of pyrrolopyrimidine-based CSF1R inhibitors. The goal of the research as well as the synthetic strategies are very clearly described, where the second one resulted in the desired compound. The screening of catalytic conditions of the Suzuki-Miyaura catalytic coupling is focused on various compounds of palladium, all with recognized ligands that already have numerous applications and are commercially available (this represents an advantage in this case). The amination step was screened using the palladium acetate and RuPhos ligand, showing high yields of the final 7-azaindole derivatives. The challenge of a final deprotection step is clearly discussed. The synthetic and catalytic protocols are straightforward and clear, the supporting data contain exhaustive characterization of new compounds. The IC50 values for CSF1R inhibition were determined. The manuscript is well written and can be recommended for publication. This referee has only one minor comment: the Authors have shown the formulae of certain ligands used in the catalytic study (Scheme 2), but some abbreviations remain undisclosed (e.g. “dba”, which is, presumably, dibenzylideneacetone). Although such abbreviations may be common in a certain area of chemistry, it would be much better for readers to have all them clearly disclosed in the main text.

Author Response

Comment 1: Although such abbreviations may be common in a certain area of chemistry, it would be much better for readers to have all them clearly disclosed in the main text

Response 2: The full name of Pd2(dba)3has been spelled out first time used (line 120). On purpose, we have not included the name of the other catalyst ligand systems in the main text, since they are so complicated, making reading of the sentences difficult. Instead they are inserted as table notes.

Reviewer 3 Report

Comments and Suggestions for Authors

The authors have presented synthetic routes to access substituted pyrrolopyrimidines via sequential Suzuki-Miyaura coupling and Buchwald-Hartwig amination. Through a systematic investigation of the reaction sequence, it was found that regioselective Suzuki coupling followed by amination is the optimal synthetic pathway. The use of an appropriate N-protecting group was crucial in achieving high reactivity. Additionally, the authors explored SEM deprotection conditions to obtain the final target compound. The azaindole was evaluated as a CSF1R kinase inhibitor to assess its bioactivity.

While this represents a significant scientific achievement, the manuscript would benefit from further refinement. Therefore, I recommend reconsideration of the manuscript after major revisions. I have outlined several suggestions below, which may help to improve the quality of the manuscript.

Top of Form

1.       The manuscript would benefit from a thorough language review, as there are multiple minor grammatical errors and the use of somewhat informal terms, such as "Quite unfortunate" in the Abstract. It is recommended to carefully proofread the entire manuscript.

2.       The abstract needs greater clarity. In its current form, it is difficult to discern the main points. For instance, providing more details on strategies, such as the chemoselective coupling of di-halide pyrrolopyrimidine, would help readers better understand the work.

3.       Please specify what kind of diseases that the FDA-approved drugs mentioned on page 1, line 34, were developed for.

4.       It would be helpful to include a ChemDraw scheme for the contents described on page 2, lines 47-56, along with the ChemDraw scheme of "this work." This would allow readers to clearly see what has been done and what represents a new approach in this manuscript.

5.       Please provide detailed reaction conditions, including reagent equivalents, in Table 1.

6.       In Table 1, please adjust the superscript format from a), b), c), d) to 1, 2, 3, 4 for consistency.

7.       On page 6, there is a second "Table 1." Please renumber this table, and note that the hydrogen atom is missing from the secondary amine.

8.       In Scheme 5 on page 6, there is a typo in entry (e). Kindly revise "CH2OOSEM" to "CH2OSEM."

9.       Regarding SEM deprotection, were TBAF conditions tested for global deprotection? If so, please provide the results.

Comments on the Quality of English Language

The manuscript would benefit from a thorough language review. Please see the comment 1.

Author Response

Comment 1: The manuscript would benefit from a thorough language review, as there are multiple minor grammatical errors and the use of somewhat informal terms, such as "Quite unfortunate" in the Abstract. It is recommended to carefully proofread the entire manuscript.

Response 1: This informal term has been removed from the abstract and in the main part of the manuscript. Some spelling errors and minor modifications to simplify the text have also been done.

Comment 2: The abstract needs greater clarity. In its current form, it is difficult to discern the main points. For instance, providing more details on strategies, such as the chemoselective coupling of di-halide pyrrolopyrimidine, would help readers better understand the work.

Response 2: we have attempted to improve the abstract as suggested.

Comment 3: Please specify what kind of diseases that the FDA-approved drugs mentioned on page 1, line 34, were developed for.

Response 3:  this has been included (treatment of tenosynovial giant cell tumours and late-stage melonoma)

Comment 4: It would be helpful to include a ChemDraw scheme for the contents described on page 2, lines 47-56, along with the ChemDraw scheme of "this work." This would allow readers to clearly see what has been done and what represents a new approach in this manuscript.

Response 4: Instead of drawing out these reactions, we decided to add short explanatory text to aid the reader. We discovered that the reference for structure II was wrong. We have included the correct reference. A reference was also inserted for structure V.

Comment 5: Please provide detailed reaction conditions, including reagent equivalents, in Table 1.

Response 5: More details has been inserted. In the original version of the manuscript, three experiments (entry 5-7) were run at 90 dgC, while most other was done at 80 dgC. Since we in this case had available starting material, we repeated these three experiments at 80 dgC. The product distribution is therefore slightly changed. The largest change in product amount was seen for the used of (Dppf)PdCl2 going from 69 to 79 mol%.  Lower temperature also led to somewhat lower amount of oxidation of the benzylic alcohols.

Comment 6: In Table 1, please adjust the superscript format from a), b), c), d) to 1, 2, 3, 4 for consistency.

Response 6: corrected

Comment 7: On page 6, there is a second "Table 1." Please renumber this table, and note that the hydrogen atom is missing from the secondary amine.

Response 7: corrected

Comment 8: In Scheme 5 on page 6, there is a typo in entry (e). Kindly revise "CH2OOSEM" to "CH2OSEM."

Response 8: corrected

Comment 9: Regarding SEM deprotection, were TBAF conditions tested for global deprotection? If so, please provide the results.

Response 9: We did not test TBAF on this structure. The following text has been added:

The use of BF3-OEt2 in the deprotection of 13d-e was also attempted, but this resulted in even more complex reaction mixtures. Tetrabutylammonium fluoride was not evaluated

Round 2

Reviewer 3 Report

Comments and Suggestions for Authors

The authors addressed the comments provided during the first revision and the quality of manuscript has been improved. There is a one minor suggestion for the current manuscript. I recommend the publication of this manuscript after editing a minor point.

- Please and 'reaction' after 'Heck' on page 2, line 54.